# Asthma Hospital Admission and Readmission Spikes, Advancing Accurate Classification to Advance Understanding of Causes

**DOI:** 10.3390/diagnostics12102445

**Published:** 2022-10-10

**Authors:** Mehak Batra, Bircan Erbas, Don Vicendese

**Affiliations:** 1Department of Public Health, School of Psychology and Public Health, La Trobe University, Melbourne, VIC 3086, Australia; 2Faculty of Public Health, Universitas AirLangga, Surabaya 60115, Indonesia; 3Violet Vines Marshman Centre for Rural Health Research, La Trobe University, Bendigo, VIC 3550, Australia; 4The Melbourne School of Population and Global Health, University of Melbourne, Carlton, VIC 3053, Australia; 5School of Engineering and Mathematical Sciences, La Trobe University, Bundoora, VIC 3053, Australia

**Keywords:** asthma, hospital, admissions, readmissions, statistical method, children

## Abstract

Background: An important component of asthma care is understanding potential causes of high asthma admissions (HAADs) or readmissions (HARDs) with potential of risk mitigation. Crucial to this research is accurately distinguishing these events from background seasonal changes and time trends. To date, classification methods have been based on ad hoc and untested definitions which may hamper understanding causes of HAADs and HARDs due to misclassification. The aim of this article is to introduce an easily applied robust statistical approach, with high classification accuracy in other settings—the Seasonal Hybrid Extreme Studentized Deviate (S-H-ESD) method. Methods: We demonstrate S-H-ESD on a time series between 1996 and 2009 of all daily paediatric asthma hospital admissions in Victoria, Australia. Results: S-H-ESD clearly identified HAADs and HARDs without applying ad hoc classification definitions, while appropriately accounting for seasonality and time trend. Importantly, it was done with statistical testing, providing evidence in support of their identification. Conclusion: S-H-ESD is useful and statistically appropriate for accurate classification of HAADs and HARDS. It obviates ad hoc approaches and presents as a means of systemizing their accurate classification and detection. This will strengthen synthesis and efficacy of research toward understanding causes of HAADs and HARDs for their risk mitigation.

## 1. Introduction

The prevalence of asthma exacerbation emergency department (ED) visits and then subsequent admission is still high among children and adolescents [1] and they create a substantial burden for children, their families and the hospital system. Particularly, the increase in early readmission within 28 days [2] is dependent on factors that we are yet to identify fully. Environmental factors have been implicated with paediatric asthma admissions. Seasonality is an important marker of total environmental load or triggers, such as high pollen exposure and respiratory virus infections, which are associated with asthma hospital admissions [2,3,4,5]. 

Methodologically, in identifying high asthma admissions days (HAADs), we are undertaking the non-trivial task of detecting anomalous points in time series which are subject to seasonality, time trends and random variation. Accurate detection is important otherwise misclassification will distort any data signals regarding possible environmental or prognostic factors. With accuracy, methodological consistency is also required so as to be able to evaluate and synthesize evidence from different studies regarding HAADs and high asthma readmissions days (HARDs) in order to provide a stronger evidence base. Accuracy and consistency in identifying HAADs and HARDs increase the potential of detecting associated risk factors whose modification may lead to an attenuation of spikes in child asthma hospital admissions and the subsequent burden on the health system. 

Anomalousness is based on the notion of occurrences that are unusual, unexpected, or, in statistical terms, extremal or outliers. That is, an unusually high spike in daily admissions time series. These terms capture anomalies on a global scale and inherent in them are ideas of distributional location and dispersion which informs the methods that have been employed to identify HAADs to date. Two studies applied a smoothing method to calculate a moving average, then used the magnitude of the residual (the difference between the average and the actual observed daily count) to determine if that day met the criterion of an HAAD. The criterion was relative to the residual standard deviation (SD) and if the residual was greater than a certain number of SDs, then that day was classified as an HAAD. One study applied a Fourier transform filter, as a way of determining the seasonally changing average and used an a priori chosen threshold of 1.96 SD [6], an a priori global (one size fits all) criterion. The second calculated a rolling average and SD based on a 25% trimmed mean, that is, only the middle 50% of the data were used, and applied a threshold of 4.5 SD that was chosen by inspecting residual quantile-quantile (qq) plots to detect a critical departure point of the large residuals from the preceding ones [7]. The second method that has been employed in past studies was model based, where a time series statistical model was applied to the data and, similarly to the previous method, the magnitude of the residual from the model predicted mean was assessed against the priori chosen threshold of 4 SDs from the mean [8]. As far as we can tell, these are the only methods that have been used for asthma admissions.

These approaches have some important limitations. The mean and SD are strongly affected by outliers. This is especially so for the SD due to its definition based on the squared distance from the mean. Hence, any definitions based on a mean and SD will tend to mask outliers when outliers are used in their calculation. Using a trimmed mean is a well-known method for reducing the effect of outliers in the calculation of the mean [9], however, excluding 50% of the data runs the risk of over smoothing, drastically restricting access to information in the data and therefore limiting sensitivity to account for seasonality and time trend in a time series. Furthermore, the use of 1.96, 4 or 4.5 SDs is not based on any validation testing to understand the impact of these definitions on sensitivity or positive predictive value in classifying HAADs. In addition, these methods do not include any formal statistical testing in regard to their classification of HAADs. They are based on the untested assertions that there are an unknown number of outliers, and they exist beyond a certain number of SDs from a sample or model predicted mean.

Anomalousness can also carry the idea of unusual or unexpected on a local scale. A high number of daily admissions for a particular time of the year may not be considered high in another, that is, it is important to account appropriately for seasonality. Similarly, a high day in one year may not be considered high in another and therefore it is also important to account for time trend. It has been shown that moving average techniques tend to filter out seasonal anomalies [10]. A time series statistical model with appropriate specification can adjust for seasonality and time trend. The time trend can be modelled in both the long term and short term, for example day of the week effects on hospital usage [11]. The limitation with model-based methods is that we are faced with model assumptions, choice, specification, and importantly, model capacity for capturing data trends. For example, in the model-based method discussed above, a log linear auto regressive statistical model was employed that accounted for seasonality and long-term time trend [8] but choices had to be made regarding log transformation and linear or non-linear specification for example. More importantly, this study made a choice of using 4 SDs as a threshold to classify HAADs. It may be asked: why not use 4.5 or 3.5? These values have not been tested regarding their sensitivity or specificity to detect HAADs.

There is a new method, published in 2017, that overcomes the critical limitations of current methods. It is termed the Seasonal Hybrid Extreme Studentized Deviate (S-H-ESD) method [10]. Validation testing of this method has shown it to be sensitive in detecting anomalous data observations both on a global and local scale, is model free and it incorporates statistical testing. Validation has shown it to have a sensitivity of about 96% and a positive predictive value (PPV) of 100% when it was applied with a statistically significant level of 0.05 in the setting of detecting anomalies in cloud infrastructure data [10]. Machine learning (ML) is in demand and has been widely used in respiratory studies, for example, COVID-19 [12,13] and COPD [14]. However, the method we put forward is straight forward to apply, does not require intensive resources and is easily understood and can be interpreted. Furthermore, our demonstrated method comes with robust statistical testing [15].

In this study, we demonstrate the use of the S-H-ESD method, a novel approach, in the important task of detecting HAADs and HARDs. We also compare it to the methods mentioned above [7,8].

## 2. Materials and Methods

### 2.1. Design/Setting

We used all Victorian private and public hospitals data obtained from the Victorian Admitted Episodes Data set (VAED) and extracted daily counts of all hospital admissions for asthma from July 1st, 1996, to June 30th, 2009, 13 years or 4748 days in total. Victoria is a state in south-eastern Australia. Only primary admissions for children (2–18 years) with a principal diagnosis of asthma of asthma (ICD-9 codes (493) up to 1998 and ICD-10 codes (J45 or J46)) were included in the study. Readmissions were defined as a subsequent admission within 28 days of the index admission discharge [11]. The time series contained 53,156 admissions including 2401 readmissions [2].

The study was commenced after obtaining ethics approval from approval from La Trobe University Human Research Ethics Committee (HEC18307).

### 2.2. Statistical Method

We define robust in the usual statistical sense as being resistant to outliers in the calculation of location and spread.

We briefly describe the new method but supply more details in the Appendix A. The S-H-ESD method relies on robust measures of location and dispersion via the median and scaled median absolute deviation (MAD). Firstly, the time series is decomposed into its trend and seasonal components using locally estimated scatterplot smoothing (STL) with an added weighting scheme to make it more robust and the residuals (the remainder) are extracted [16]. The residuals are passed to the Rosner Extreme Studentized Test (ESD) [17]. The ESD uses a statistical test based on the null hypothesis that there are no outliers against the alternative that there are up to k outliers, where k is chosen by the user. The level of statistical significance can be chosen as required and is subject to Bonferroni correction based on the number of detected outliers. The test iterates through the data, removing the found anomaly for the next iteration. Choice of k can be adjusted until beyond which, no further outliers are detected and hence it is an exhaustive method. The ESD was initially formulated using the sample mean and SD and requires approximate normality as it refers to a t-distribution. Within S-H-ESD, the sample mean and SD are replaced by the median and scaled MAD, robust measures of location and dispersion, respectively [18,19], and robustness is augmented by the use of a robust weighting scheme for extracting the residuals from the time series. This decomposition facilitates S.H.ESD to detect global and local anomalies and ensures that the residuals have a unimodal distribution which makes the choice of the ESD appropriate [10]. For further details regarding the metrics to evaluate this method, please refer to Appendix A.

We compare S-H-ESD to two other methods previously used for identifying days of unusually high asthma admissions.

1.Similarly, to the model-based approach by Newson et al. [8], we used a semi parametric general additive model (GAM) [20] to model mean asthma admission and readmission daily counts, adjusting for seasonality, time trend and day of week effect as done previously with these VAED data [11]. In line with Newson et al., we used the a priori definition of a residual being 4 SD from the model predicted mean as a threshold to identify HAADs and HARDs. We refer to this method as M.4SD, where M signifies model based.2.We follow the example of Silvers et al. [7] and use a rolling 25% trimmed mean and SD then choose a threshold based on the inspection of residual qq plots. We refer to this method as TMQQ (trimmed mean qq plot).

We compare if the identified HAADs and HARDs are reasonable according to what may be expected from what is known about the seasonality and time trends of asthma admissions and readmissions in Victoria from our past research [2,11]. For time trend, we compare the number of HAADs and HARDs to pre and post 2002 as child asthma hospital readmissions reduced from 1997 to 2002 but showed an increasing trend to 2009 and admissions reduced and then flattened out from about 2002. It would not be expected that HAADs and HARDs follow seasonality and time trends completely, by definition they are anomalous, but it would be expected that their likelihood would increase when more admissions occur, and it is well known that there is a strong seasonal aspect to child asthma hospital admissions. We present tables for the seasonality and time trend results. We also make comparisons of the days selected as HAADs and HARDs by the three methods in context of the time series themselves, for which we present graphical evidence. Our comparisons are basically descriptive although we did conduct some simulations, see Appendix A. The S.H.ESD has already been subject to comprehensive validation testing for its application to cloud computing and we wish to compare methods used for the study of asthma hospital admissions as a way of alerting the asthma research community to this method. The methods were implemented with freeware R [21]. S.H.ESD was implemented via the AnomalyDetection library [22] and a statistical significance level of *p* < 0.05 was nominated in classifying HAADs and HARDs. The R libraries mgcv [23], ggplot2 [24] and stlplus [25] were used for the GAM model, graph plotting and time series decomposition, respectively. We also supply an R computer script for this method, see Appendix A.

## 3. Results

Daily admission counts ranged between 0 and 51 (mean 11.3, SD 6.0). Daily readmission counts ranged between 0 and 5 (mean 0.5, SD 0.7) and only 15 (0.3%) and 2 (0.04%) days had daily readmissions of 4 and 5, respectively. See Appendix A where we demonstrate STL decomposition [16] and which show the admissions and readmissions time series and their three components of time trend, seasonal fluctuation and the remainder (residuals). The seasonal and trend components have noticeable effects on both time series. As expected from our previous research, they show that the long-term time trend had been a decrease in admissions to about 2002 followed by a largely flat period but with a little oscillation and that readmissions also decreased to 2002 but was followed by an increasing trend to study period end [2,11].

In applying TMQQ, we found that the qq plots indicated thresholds of 10.2 and 7.5 SDs to identify HAADs and HARDs, respectively. The results of applying the three methods of S.H.ESD, M.4SD and TMQQ to all daily admissions and readmissions are displayed in Table 1 by month of occurrence to display their seasonality and Table 2 to describe time trends relative to pre and post 2002.

### 3.1. High Asthma Admission Days (HAADs)

#### 3.1.1. S-H-ESD

Seventeen days (0.4%) were classified as HAADs (*p* < 0.05) and they had between 33 and 51 daily admissions, see Figure 1. The most frequent month of occurrence was February (summer end and return to school) with 10 (59%), followed by May (autumn end) with 3 (18%). November (mid pollen season) had two HAADS. These months are consistent with seasonal peaks in child asthma admissions as shown from our previous research [2,11]. Seven of the months did not register any HAADs. This method detected more HAADs pre 2002 compared to post 2002 which reflects the long-term time trend in the data.

#### 3.1.2. TMQQ

Twenty-three days (0.5%) were classified as HAADs and they had between 14 and 51 daily admissions, see Figure 1. The most frequent month of occurrence was February (summer end and return to school) with 20 (87%) followed by April, May and November with 1 (4%) each. These months are consistent with seasonal peaks in child asthma admissions. The remaining eight months did not have any days classified as HAADs. This method detected many less HAADs pre 2002 compared to post 2002, 26% compared to 74%, respectively, which is not consistent with the long-term time trend in the data.

#### 3.1.3. M.4SD

Seven days (0.2%) were classified as HAADs and they had between 28 and 51 daily admissions, see Figure 1. Four (57%) of the HAADs occurred in November (mid pollen season) followed by February (summer end) and March (autumn start) with 2 (29%) and 1 14%), respectively. Although they are small numbers, this distributional spread does not seem consistent with child asthma hospital admission in Victoria as a greater percentage is expected in autumn compared to spring [2]. Pre and post 2002 comparisons were consistent with known time trends.

### 3.2. High Asthma Readmission Days (HARDs)

#### 3.2.1. S-H-ESD

In applying this method, we found that it failed for the detection of HARDs in our data set. It classified 39.4% of the readmissions as anomalous, many of which were daily counts of 1 or 2, a spurious result given the meaning of outlier. This was mainly due to the child asthma hospital readmission time series being a low count series with a range of 0–5, that is, highly discrete and was dominated by zero (60th percentile). If more than 50% of values are the same, then the MAD will equal zero and the method breaks down. We overcame this problem by adding smoothness using random noise from a uniform distribution between, but not including, −0.5 and 0.5. Our simulation testing was based on the addition of smoothness. See Appendix A for more details.

After the addition of smoothness, there were 25 days (0.5%) classified as HARDs (*p* < 0.05) and they ranged between three and five daily readmissions, see Figure 2. All of the days with four and five readmissions (highest) and eight of the days with three readmissions were classified as HARDs. The most frequent month of occurrence was August (winter end) with seven (28%) followed by June (winter start) six (24%). These months are consistent with seasonal peaks in child asthma readmissions. All summer months and July (mid-winter) did not have any HARDs. More HARDS occurred post compared to pre-2002, which is consistent with the long-term time trend.

#### 3.2.2. TMQQ

Twenty-three days (0.5) were classified as HARDs and they ranged between two and five daily readmissions, see Figure 2. Only one of the two days with five readmissions and three of the fifteen days with four readmissions were classified as HARDs. The months of most frequent occurrence were February (summer end and return to school), March and October (pollen season start) with five each (23%). February and October are not consistent with child asthma hospital readmission peaks in these data [11]. One HARD was classified for January when readmissions are historically very low. More HARDs were classified post 2002, but the difference compared to pre 2002 was close to an even split, 10 compared to 12, much less than the other two methods. This led us to consider this result as not consistent with the long-term time trend.

#### 3.2.3. M.4SD

Eighteen days (0.4%) were classified as HARDs, and they ranged between three and five daily readmissions, see Figure 2. This method classified all the days with four or five readmissions and one of the days with three as a HARD. The most frequent month of occurrence was June (winter start) followed by March (autumn start) with five occurrences. These months are consistent with seasonal peaks in child asthma readmissions. Pre and post 2002 comparisons were consistent with time trend.

These comparisons of results are summarized in Table 3.

## 4. Discussion

In this study, we demonstrated the S-H-ESD method, an alternative robust technique to detect HAADs and HARDs, and compared it to two previously used methods for asthma admissions. We found less HAADs but more HARDs after 2002, which possibly was due to instability in the readmissions time series post 2002. That is, despite an overall lower number of admissions compared to pre-2002, a higher number of anomalous readmission days were identified post 2002. We showed how to extend S.H.ESD in the situation where the MAD equals zero. There were clear differences between the results obtained from the three methods. For HAADs, Figure 1 indicates that S.H.ESD classified the days that would be expected to be classified as HAADs indicating good sensitivity or low false negatives and had not classified days that would be expected not to be classified as HAADs (good PPV or low false positives). Whereas the TMQQ and M.4SD methods both missed some obviously high days (false negatives) and TMQQ classified many lower days, as low as 14 admissions, as HAADs (false positives). In the context of seasonality and time trend, comparing to other days close by, these low days classified by TMQQ could not be reasonably defended as HAADs as the mean admission count was 11.2. M.4SD did not seem to be prone to false positives as it mainly classified days with higher counts, 30 or above, but did classify two days with counts of 28 and 29 which are on the edge of credibility considering the many more days with higher counts. However, in context of the much lower counts in nearby days, these two days may be defensible. M.4SD had the lowest classification rate for HAADs, about half or less than the other two methods. It did not classify many of the high days that would be expected to be classified indicating a lower sensitivity (false negatives). From Figure 1, it is interesting to note that there is little corroboration between the three methods. Of the 38 distinct days that were classified as HAADs by the three methods, only 3 days were chosen by all three methods and 3 days by two methods. S.H.ESD figured in all those corroborations indicating it likely had greater sensitivity than the other two methods.

For HARDs, Figure 2 indicates that S.H.ESD and M.4SD performed equally well. They both chose all the very high days of four or five readmissions and a few of the days with three readmissions but on which they corroborated on one of them only. In contrast, TMQQ classified only one of the two days with five readmissions and only 3 of the 15 days with four readmissions indicating a low sensitivity, or propensity for false negatives. TMQQ also chose 11 days with only two readmissions, which in context of this very low-count time series would be difficult to defend and indicated low PPV, a propensity for false positives.

TMQQ’s difficulty with both the admission and readmission time series was likely due to a combination of its two main features. Its strong filtering mechanism of using only the middle 50% of the data to calculate a SD (moving) would have the effect of decreasing its magnitude because of reduced data variation. This increases the likelihood of false positives because distances from the mean would seem relatively larger in units of a smaller SD. It has also been shown that use of a moving average tends to hide seasonal anomalies and hence may make TMQQ prone to false negatives [10]. TMQQ also has the limitation of an ad hoc choice of trimming width. It may be asked: why choose 25%, why not 15%? It is not clear, what affect this might have on model sensitivity or PPV. We also found that choice of threshold criterion when assessing the residual qq plot could be subjective and difficult. It was not completely clear where to locate a critical departure point of the large residuals from the preceding ones [7].

M.4SD seemed to perform well with HARDs. This was likely due to the selection of 4 SD as the threshold criterion which happened to work well with the model we had chosen. The GAM we used was chosen because we understood its good performance in past research with these low count time series data [11]. However, this combination did not prove as serendipitous in the classification of HAADs as M.4SD seemed to be hampered by both false positives and false negatives. The limitation of M.4SD hinges on the need for model development, with all the choices that go with it, to account for data variation in order to make accurate predictions. After which, a choice of criterion for the number of SDs needs to be made in the presence of uncertainty about the effect on classification sensitivity and PPV.

In contrast, S.H.ESD was consistent in identifying HAADs and HARDs. From graphical evidence, it classified days as HAADs or HARDs that would be expected to be classified and did not classify days that would be expected not to be classified. The seasonality and time trends of the classified HAADs and HARDs, as best could be assessed with small numbers, also corresponded to the seasonality and time trends of the underlying asthma admissions and readmissions. The S.H.ESD method was able to classify HAADs and HARDs without imposing an a priori or ad hoc definition of a high day as used by M.4SD or a data driven definition as done with TMQQ. In contrast to both TMQQ and M.4SD, S-H-ESD provided statistical evidence for the identification of HAADs and HARDs which the two other methods do not provide. S.H.ESD was easy to implement, as can be seen from the provided R computer code, see Appendix A. The adding of smoothness, if required, is also straight forward to implement.

Although it worked well with our data, the developers of S.H.ESD felt its capacity to capture long-term trend needed to be developed further [10]. This is important to minimize false positives and is the subject of further research. In saying that, it would be useful to test and validate our method in data sets from many different countries as S.H.ESD has the potential to standardize and synthesize similar research globally.

S.H.ESD presents as a suitable method to accurately identify HAADs and HARDs which would support research on these phenomena by reducing misclassification error due to false positives and false negatives. This is a crucial consideration for understanding the causes of HAADs and HARDs. If we seek to understand factors that are associated with high admission or readmission days, we must be as accurate as possible to identify them or we risk distorting any signal in the data because of misclassification. The application of different ad hoc definitions for HAADs by different studies, makes comparison of study results difficult. Because of this, synthesis of study results in order to promote understanding of causes of HAADs and HARDs is hindered. Because the S-H-ESD method works identically in any data set without any ad hoc or a priori definitions for a HAAD or HARD, this source of heterogeneity between different studies would be removed which would also raise the potential of promoting their synthesis.

This study has the strength of using a comprehensive data set of two time series of 13 years in length with which to compare the three methods. The limitation of our study is that the basis of the comparisons was graphical and descriptive and was not based on simulated data sets with known outcomes. However, the S.H.ESD method has been internally validated previously and shown to have a sensitivity and PPV of 96% and 100%, respectively, at the 0.05 level of statistical evidence [10]. The other two methods have never been tested in this way. Nevertheless, the aim of this article was to demonstrate the method, not to validate it.

## 5. Conclusions

The Seasonal Hybrid Extreme Studentized Deviate (S.H.ESD) method is easy to use and seems accurate in the identification of high asthma admission and readmission days. In contrast to other methods, S.H.ESD supplies appropriate statistical evidence for the identification of high admission days. Although we demonstrated the method on a paediatric asthma hospital admission data set, it can also be applied to adult asthma admissions or other time series in general.

S.H.ESD obviates the need for a priori classification criteria or ad hoc modelling and so promotes consistency and accuracy of research. It also presents as a means of systemizing the identification of days of high child asthma hospital admissions and readmissions. Consequently, this may have the benefit of opening up the potential of synthesizing research in this area from many groups across the globe. However, further study is required to corroborate the effectiveness of S.H.ESD for the accurate identification of days of high child asthma hospital admissions and readmissions.

## Figures and Tables

**Figure 1 diagnostics-12-02445-f001:**
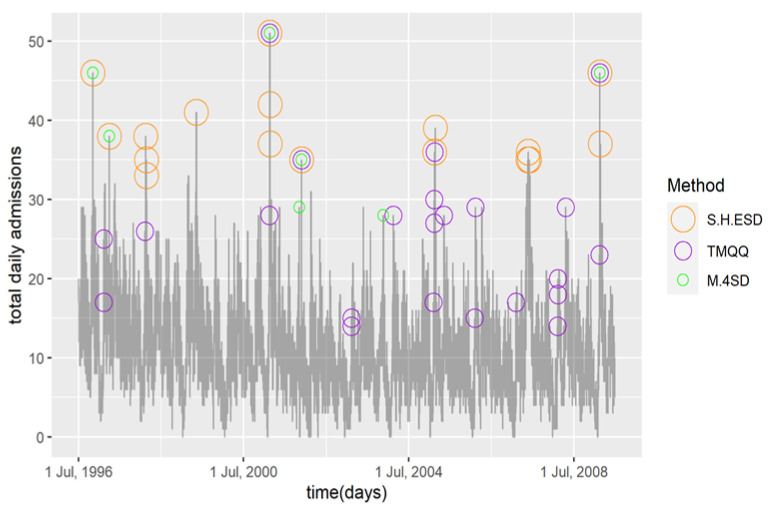
Time series of daily child asthma hospital admissions in Victoria with HAADs classified by the three compared methods.

**Figure 2 diagnostics-12-02445-f002:**
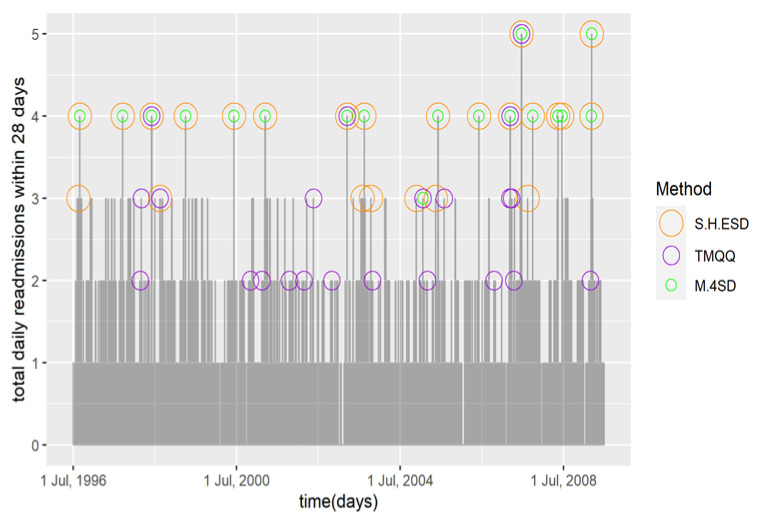
Time series of daily child asthma hospital readmissions within 28 days in Victoria with HARDs classified by the three compared methods.

**Table 1 diagnostics-12-02445-t001:** Number of days classified as high asthma admission (HAADs) and high asthma readmission (HARDs) daily counts with the three reviewed methods by month of occurrence over the 13 years of the study period.

Month *	HAAD			HARD		
	S.H.ESD ^†^	TMQQ ^‡^	M.4SD ^⁋^	S.H.ESD ^†^	TMQQ ^‡^	M.4SD ^⁋^
**December**	0 (0%)	0 (0%)	0 (0%)	0 (0%)	0 (0%)	0 (0%)
**January**	0 (0%)	0 (0%)	0 (0%)	0 (0%)	1 (4.5%)	1 (5.6%)
**February**	10 (59%)	20 (87%)	2 (29%)	0 (0%)	5 (22.7%)	0 (0%)
**March**	1 (6%)	0 (0%)	1 (14%)	5 (20%)	5 (22.7%)	5 (27.8%)
**April**	0 (0%)	1 (4%)	0 (0%)	1 (4%)	1 (4.5%)	1 (5.6%)
**May**	3 (18%)	1 (4%)	0 (0%)	2 (8%)	1 (4.5%)	1 (5.6%)
**June**	1 (6%)	0 (0%)	0 (0%)	6 (24%)	2 (9.1%)	6 (33.3%)
**July**	0 (0%)	0 (0%)	0 (0%)	0 (0%)	1 (4.5%)	0 (0%)
**August**	0 (0%)	0 (0%)	0 (0%)	7 (28%)	1 (4.5%)	2 (11.1%)
**September**	0 (0%)	0 (0%)	0 (0%)	2 (8%)	0 (0%)	2 (11.1%)
**October**	0 (0%)	0 (0%)	0 (0%)	1 (4%)	5 (22.7%)	0 (0%)
**November**	2 (12%)	1 (4%)	4 (57%)	1 (4%)	0 (0%)	0 (0%)
**Total**	17 (101%)	23 (99%)	7 (100%)	25 (100%)	22 (99.7%)	18 (100.1%)
**Total as % of 4748 Days**	0.4%	0.5%	0.2%	0.5%	0.5%	0.4%

* December is the start of summer. Pollen season starts October through to December. † Seasonal Hybrid Extreme Studentized Deviate test (see methods section). ‡ Using the method of a 25% trimmed mean (middle 50% of the data) and quantile-quantile plots to choose the number of SD a positive residual is from the mean to define an unusually high count [7]. ⁋ 4 standard deviations for a model positive residual to be from the predicted mean as a priori definition of an unusually high count.

**Table 2 diagnostics-12-02445-t002:** Number of days classified as HAAD or HARD comparing study years pre and post 2002.

Year	HAAD			HARD		
	S.H.ESD	TMQQ	M.4SD	S.H.ESD	TMQQ	M.4SD
**<= 2002**	10 (59%)	6 (26%)	5 (71%)	9 (36%)	10 (45%)	6 (33%)
**> 2002**	7 (41%)	17 (74%)	2 (29%)	16 (64%)	12 (55%)	12 (67%)
**Total**	17 (100%)	23 (100%)	7 (100%)	25 (100%)	22 (100%)	18 (100%)

**Table 3 diagnostics-12-02445-t003:** Summary of method consistency with seasonality, time trend and size of HAADs and HARDs.

Year	HAAD			HARD		
	S.H.ESD	TMQQ	M.4SD	S.H.ESD	TMQQ	M.4SD
**Seasonality**	Yes	Yes	No	Yes	No	Yes
**Time trend**	Yes	No	Yes	Yes	No	Yes
**Size**	Yes	No	Yes	Yes	No	Yes

## Data Availability

Hospital admission records are restricted by The Department of Health and Human Services, Victoria (DHHS), and therefore we cannot make the data publicly available. However, qualified researchers may apply to DHHS for access. For further information, and data request, please see https://www.health.vic.gov.au/reporting-planning-data/for-researchers.

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
