# Peer review of "Asthma Hospital Admission and Readmission Spikes, Advancing Accurate Classification to Advance Understanding of Causes"

_diagnostics, 2022, doi:10.3390/diagnostics12102445_

Round 1

Reviewer 1 Report

This is an overall interesting study, providing an approach for detection of outliers in a time series context. I have only some suggestion for improving the results' presentation and the information reported in the manuscript.

Major comments:

1.      Please provide a better description of the assessed dataset. Does it contain all hospital admissions in public and private hospitals from the State of Victoria? How are hospital admissions defined in that dataset? In addition, is there any reason on why analysing data only up to 2009?

2.      When comparing the three approaches to identify HAADs and HARDs, you analysed whether more HAADs/HARDs occurred before or after 2002, given that overall child hospital admissions and readmissions decreased up to 2002 and then increased or flattened. However, it is possible that after 2002 there were less admissions is most days, but a higher number of days with an outlier number of admissions (i.e., a “a more unstable pattern”, but associated with an overall lower number of admissions). I would discuss that possibility on the Discussion section.

3.      In the discussion, you occasionally refer to the “sensitivity”, “predictive values”, “false positives” and “false negatives” of the methods you apply. However, such had not been formally computed and no results are provided for those properties in the Results section. In addition, it is referred that “the limitation of our study is that the basis of the comparisons was graphical and descriptive”. However, in the methods and in the results sections, we have no description of a formal graphical comparison. I understand that there is no better gold-standard and that graphical comparisons were the ones you were able to perform. However, I would perform such comparisons in a more formal way. That is, I would have a graphical classification of each day (as an HAAD vs no HAAD) by two blinded independent observers and I would compare the results of such classification by the results of each model (so that I would be able to compute specificity, sensitivity, predictive values and agreement measures).

4.      In the discussion, you state that “From figure 1, it is interesting to 292 note that there is little corroboration between the three methods.”. You may calculate agreement measures between the three methods you applied (e.g., kappa coefficient).

Minor comments:

1.      In the last paragraph of the introduction, you state that "Validation testing of 100 this method has shown it to be sensitive in detecting anomalous data observations both 101 on a global and local scale, is model free and it incorporates statistical testing. Validation 102 has shown it to have a sensitivity of about 96% and a positive predictive value (PPV) of 103 100% when it was applied with a statistically significant level of 0.05 in the setting of de-104 tecting anomalies in cloud infrastructure data". The last paragraph of the Introduction is usually where the aim of the manuscript is stated. The aforementioned sentence may confuse the reader on whether that is a result from your study or information from the literature.

2.      In the sentence “Firstly the time series is decomposed into its time and seasonal components” (line 124), did you mean “…its trend and seasonal components”?

3.      The figures S1 and S2 have the same legend. Can you please provide more information on the difference between them?

Reviewer 2 Report

Dear Sir,

The paper is well written. Please do the following changes. I have no hesitation in recommending this manuscript once the changes are addressed.

1. In introduction section you need to mention the research gaps and objectives

2. Also mention the novelty in your research.

3. Add literature review, Machine learning is being used to combat dangerous diseases such as COVID-19 and monkey pox. Include these papers.

1. Supervised Learning Models for the Preliminary Detection of COVID-19 in Patients Using Demographic and Epidemiological Parameters

2. Sv P, Ittamalla R. What concerns the general public the most about monkeypox virus?–A text analytics study based on Natural Language Processing (NLP). Travel Medicine and Infectious Disease. 2022 Jul 31.

3. Zhang F. Application of machine learning in CT images and X-rays of COVID-19 pneumonia. Medicine. 2021 Sep 9;100(36).

Review a few more papers where machine learning is in deep demand

4.  Explain the evaluation metrics

5. Add a comparison of similar researches. 

6. Add the challenges and future directions.

Paper is good. Do the following changes and resubmit.
